# BMI Alterations and Prevalence of Overweight and Obesity Related to Service Duration at the German Armed Forces

**DOI:** 10.3390/healthcare11020225

**Published:** 2023-01-11

**Authors:** Lorenz Scheit, Barbara End, Jan Schröder, Manuela Andrea Hoffmann, Rüdiger Reer

**Affiliations:** 1Bundeswehr Hospital Hamburg, Clinic I—Internal Medicine, Lesserstr. 180, 22049 Hamburg, Germany; 2Department of Sports Medicine, Faculty for Psychology and Human Movement Science, Institute for Human Movement Science, University of Hamburg, Turmweg 2, 20148 Hamburg, Germany; 3Institute for Preventive Medicine of the German Armed Forces, Aktienstr. 87, 56626 Andernach, Germany; 4University Medical Center of the Johannes Gutenberg-University, Langenbeckstraße 1, 55131 Mainz, Germany

**Keywords:** Bundeswehr, army, body mass index, overweight, obesity, military

## Abstract

The aim of this study was to assess body mass index (BMI) and the prevalence of overweight and obesity at entry and release of service at the German Armed Forces and related associations to service duration. In a cohort study, 85,076 paired BMI data sets (entry and release of service) of German soldiers (5.4% females) between 2010 to 2022 were analyzed retrospectively to assess BMI alterations and the prevalence of overweight (BMI ≥ 25) and obesity (BMI ≥ 30) after service durations of ≤2 years, 2–5 years, or ≥5 years. Between 2010 and 2022, we observed a trend for BMI increases of about 0.5 kg/m^2^ (X^2^ = 27.104, *p* = 0.007). BMI increases differed significantly (X^2^ = 7622.858, *p* < 0.001) after ≤2 years (0.0 kg/m^2^), after 2–5 years (1.1 kg/m^2^), and after ≥5 years (2.4 kg/m^2^) and were correlated to service duration (r = 0.34, *p* < 0.001). The prevalence of overweight increased from 33.0% to 39.5%. Obesity prevalence increased from 3.7% to 6.3%. The switch to obesity was more pronounced for longer service durations. Although secular trends for BMI increases among soldiers were in line with the general population, service duration was related to BMI increases. Especially, the service time depending on pronounced prevalence of obesity should be a matter of debate leading to counteracting measures at the German Armed Forces.

## 1. Introduction

Globally, the incidence of overweight and obesity has increased significantly. Between 1975 and 2004, the body mass index (BMI) increased from 21.7 kg/m^2^ to 24.2 kg/m^2^ on average worldwide. For the German population, especially among younger adults, an alarming trend of an increasing prevalence of obesity was observed (1998 younger adults: 18.9% males, 22.5% females; 2011 younger adults: 23.3% males, 23.9% females), while the rate of overweight individuals remained constant on a high level (67.1% men, 53.0% women) between 1998 and 2011 [1]. This development is actually identified as a highly relevant symptom because obesity and even overweight are well known as being correlated to increases of morbidity, e.g., metabolic syndrome and related cardio-vascular diseases as well as some kinds of cancer, which in turn lead to an increased all-cause mortality [2,3,4]. From a military perspective, it must not be ignored that the tendencies observed within the general population are often reflected among military service members as well. For example, a study of more than 700,000 U.S. Army soldiers between 2001 and 2011 found that 34.2% were overweight and 10.0% were obese [5]. However, the prevalence of overweight and obesity may vary in other countries, cultures, or in different military units and national armed forces, which may also depend on the decades of data analysis, e.g., USA [5], China [3], Poland [6], and UK [7].

Members of military units generally consist of younger individuals who are at the beginning of their military enlistment of employment, and these young adults generally must be in a good health and possess certain mental and physical fitness characteristics to perform their military duties in the armed forces. Maintaining military readiness therefore requires a high level of physical activity (PA) and even sport. In the German Armed Forces, the level of physical fitness was assessed annually by the so-called ‘IGF’ (Individual Basic Skills) [8]. With respect to the German Armed Forces, we found only one study published by Weber et al. who reported a prevalence of 0.9% (95% CI 0.7–1.1) for individuals with symptoms of metabolic syndrome in a large sample of 12,014 members of the German Air Force. The higher the BMI, the higher this prevalence rate of metabolic syndrome [9].

Accordingly, it may be noted that between 2016 and 2020, only 0.8% of all applicants for service at the German Armed Forces were rejected due to their obesity and assumed health and fitness limitations [10]. However, a systematic literature search in Pubmed did not identify any studies that examined BMI itself as a surrogate parameter for fitness and health, nor were any studies published on BMI increases during the period between the beginning and end of service in the Armed Forces. Recently, this topic was also questioned in the German parliament and could not be answered sufficiently [10]. Thus, the question of BMI status, its increases, and the rate of overweight or obesity is considered a relevant research gap.

Therefore, the aim of this study was to investigate the BMI situation of members of the German Armed Forces in a large-scale sample of soldiers. A key objective was to conduct analyses of BMI changes from the beginning to the end of service to identify possible associations between BMI increases and service time in the Armed Forces. In addition, a likely related shift from normal weight to overweight or obese has been suggested. However, it should be emphasized that our analyses did not intend to provide a medical risk analysis of potentially overweight- or obesity-related diseases. For those health risk analyses, the German Armed Forces switched from the assessment of measuring BMI to the waist-to-height-ratio (WHtR) in July 2018 as other forces have already done, e.g., British Army or French Armed Forces [7,10,11].

Our expectation was that baseline BMI levels would not differ significantly from those of the age-adjusted general population, and we expected rates of overweight and obesity to be no higher than in other European armed forces. However, we hypothesized that BMI increases would correlate with elapsed time from onset to end of employment, likely reflecting a disproportionate BMI increase in soldiers compared to age-matched subsamples of the German general population drawn from the literature, which in turn should lead to subsequent actions to address the likely limitations of fitness and readiness.

## 2. Materials and Methods

### 2.1. Study Design

The present study analyzed data from the years 2010 to 2022 in terms of a cohort study (level of evidence: 3).

### 2.2. Data Acquisition and Preparation

Data were provided by the Institute of Preventive Medicine of the German Armed Forces and consisted of the date of data collection and the date of birth, which were used to calculate the age at the time point of entry into service or discharge of service, which was assigned by the coded survey data. In addition, BMI (kg/m^2^) was calculated from height (cm) and body weight (kg) at the respective time points (entry into or discharge from the German Armed Forces), which served as the primary endpoint for statistical analyses and from which we calculated the proportion of overweight (BMI ≥ 25.0 kg/m^2^) or obesity (BMI ≥ 30.0 kg/m^2^) according to the World Health Organization (WHO) definition of 2000 [12].

The entire data set included 473,246 assessments (93.5% males, 6.5% females) of the German Armed Forces. These anonymous data referred to physical examinations at the beginning (*n* = 226,576; 91.8% males) and at the end (n = 246,670; 95.1% males) of the soldiers’ service at the German Armed Forces between 2010 and 2022. Of course, these numbers have to be interpreted as a dynamic equilibrium, meaning that we have included individuals who left service, for example, in 2010 or later, but started service before our observation period, and conversely others who started service in 2022 or earlier and whose employment end will be after the end of our observation. Additionally, some of the included soldiers had more than one examination when entering or leaving the service.

Thus, a data filter was applied, and the remaining sample size was substantially reduced resulting in a number of 85,076 (94.7% males) paired data sets. After an initial time-trend analysis (BMI at the entry time point from 2010 to 2022), those paired data sets were ordered by their duration of service in the German Armed Forces in order to conduct pre–post (or: entry–exit) analyses. First, we defined one-year increments based on employment days identified during beginning and end-of-service health assessment visits (Table 1).

Then, we clustered these service duration classes into 3 intervals (less than 2 years, 2 to less than 5 years, 5 and more years) (Table 2).

As primary outcome, BMI is also known to be a function of age beside others. Thus, descriptive age analyses are presented in the Section 3.

### 2.3. Statistical Methods

Testing for normality (Kolmogorov–Smirnov test) revealed significance. Due to the non-normal distributed data, non-parametric statistics were used to describe the central tendency as median (Med: 50% percentile) and the variation as inter-quartile-range (IQR: 25% to 75%). For frequency analyses, the counts (n) and their percentage (%) were reported. For contingency analysis in the case of categorical variables (e.g., overweight or obesity status at entry or end of service) we calculated chi^2^-based McNemar tests and the contingency coefficient (C), and for continuous variables (e.g., employment duration, BMI alterations), we calculated the Spearman correlation coefficient (r_s_). The non-parametric Mann–Whitney U-test was used for comparisons between non-normally distributed samples (e.g., sex differences in age), and the Wilcoxon test was used for dependent variables (e.g., BMI from entry to release of service).

The Kruskal–Wallis test was conducted to analyze differences between more than two independent samples (e.g., BMI values at the service entry years from 2010 to 2022 or BMI increases for service duration clusters) (IBM SPSS V.23, Armonk, NY, USA).

Due to our large sample sizes, the significance level was set at *p* ≤ 0.001.

## 3. Results

### 3.1. Subjects’ Age at Service Entry

As the soldiers’ age showed a non-normal right-skewed distribution, data had to be described by their median (50%) and the respective IQR (25% to 75%). At the entry to service, soldiers had an age of about 20.2 years (19.2–21.7) with males being a little older than females (males 20.2 (19.2–21.6) years vs. females 20.0 (18.7–22.4) years, Z = 4.952, *p* < 0.001). The age at the release of service depended highly on the individuals’ service time. Thus, an overall averaged analysis was not reasonable, because the whole sample was dominated by the majority of cases belonging to the service time cluster of two years or less (Table 2).

### 3.2. Subjects’ Body Mass Index (BMI)

Global analyses—irrespective of the length of time in duty—revealed a significant BMI shift of on average about 0.6 kg/m^2^ (Z = 100.247, *p* < 0.001) from the entry (23.5 kg/m^2^ (21.5–25.7)) to the release of service (24.1 kg/m^2^ (22.1–26.3)). In detail, 26,416 soldiers showed no BMI changes, while a minority of 18,462 soldiers showed BMI decreases, and a majority of 40,151 soldiers showed BMI increases.

Sex differences were irrelevant (females showed on average about 1 kg/m^2^ lower BMI values at the entry and the release of service, but represented only 5.3% of the whole sample.

BMI values at the service entry date increased slightly (trend approx. Δ + 0.5 kg/m^2^) from the year 2010 to 2022 (Chi^2^ = 27.104, *p* = 0.007) (Appendix A).

This approach describing the whole sample helps to understand the BMI state of more than 85,000 younger adults in their early twenties, and in particular the slight tendency of BMI increases of German Armed Forces personnel at their entry into service during the 12-year observation period (from 2010 to 2022), e.g., for comparisons with the general population.

Any individualized analysis of BMI increases was related to the time span of service-specific pair wise comparisons for all included cases, with their service time clustered into duration intervals (<2 years, 2 to <5 years, 5 to >12 years). Table 3 provides detailed information on BMI status at the beginning and the end of service, as well as the respective BMI changes in the above-mentioned clustered service classes.

Males demonstrated a higher BMI at the beginning as well as at the end of their employment compared to females, and BMI increases after employment durations of ‘under 2 years’, ‘2 to less than 5 years’, and of ‘5 to more than 12 years’ were even more pronounced among male soldiers. There was a significant effect for larger BMI increases the longer the service duration lasted (Chi^2^ = 7622.858, *p* < 0.001) (Table 3).

In particular, the service duration clusters differed significantly in their BMI increases one from another (all pairs, *p* < 0.001)

This ranking was associated with a significant correlation between years of service and BMI changes within the whole sample of 85,076 paired data sets (r_s_ = 0.34, *p* < 0.001).

Based on these paired data sets, we determined the prevalence of overweight (BMI > 24.9 kg/m^2^) at entry into employment to be 33.0% overall (females 25.3%, males 33.5%) and at the end as in total 39.5% (females 30.2%, males 40.1%). The respective prevalence of obesity (BMI > 29.9 kg/m^2^) at entry into service was in total 3.7% (females 1.6%, males 3.8%) and at the end 6.3% (females 5.7%, males 6.3%).

With consideration of the duration of service, the trend for the prevalence of overweight (Table 4) or obesity (Table 5) was as follows.

Within the largest subsample of soldiers with a service of less than 2 years, 6.4% started as normal-weighted individuals and became overweight during their service time (X^2^ = 40,660.344, *p* < 0.001, C = 0.61) (Table 4). Moreover, 1.8% soldiers became obese during their service (X^2^ = 27,112.948, *p* < 0.001, C = 0.54) (Table 5).

Among those with of 2 to 5 years of service, 18.3% started out normal weight and became overweight during their service in the German Armed Forces (X^2^ = 3968.344, *p* < 0.001, C = 0.51) (Table 4). Within this subsample, 8.3% started with a BMI < 30 kg/m^2^ and became obese during their service time (X^2^ = 1286.448, *p* < 0.001, C = 0.32) (Table 5).

For the smallest subsample of individuals with more than 5 years of service (n = 6557), we identified 33.1% starting out normal weight and becoming overweight during their service in the German Armed Forces (X^2^ = 1059.699, *p* < 0.001, C = 0.37) (Table 4). Within this sample 15.1% were overweight but not obese at entry into the German Armed Forces, and then experienced an increase in BMI leading to obesity by the end of their employment after 5 or more years (X^2^ = 413.384, *p* < 0.001, C = 0.24) (Table 5).

## 4. Discussion

This study aimed to describe the BMI status among soldiers of the German Armed Forces and BMI alterations as well as a shift of prevalence from normal to overweight or even obesity as a function of length of service.

As we expected to find probably disproportionate increases in BMI over time during service in the German Armed Forces, we had to rule out secular or biological aging effects. Thus, our data assessed between 2010 and 2022 should be interpreted in front of the age-adjusted BMI status of the general population.

Our key finding, based on more than 85,000 paired data sets, was a significant correlation between employment duration and BMI increases, with the highest increases of almost 2.4 kg per square meter for those who were employed five years or longer at the German Armed Forces. This BMI increase for service of more than 5 years (up to 12 years) might probably be explained—at least partly—by the biological aging effect, which was also observed in the general German population. Mensink et al. reported BMI increases of on average 1.5 kg/m^2^ for females and 2.3 kg/m^2^ for males in the German general population between the age groups of 18–29 years and 30–39 years [1]. Although our data do not allow a direct assignment to a distinct age at entry into service, a similarity may be assumed for soldiers serving longer than 5 years, even if not all average BMI increases in soldiers can be explained. Referring to 5-year increments within the German general population, BMI increases of 1 kg/m^2^ were reported between age clusters of 18–20 and 20–25 years, and an additional 1.1 kg/m^2^ increment was reported toward the age cluster 25–30 years—meaning 2.1 kg/m^2^ from the regular entry to the age of discharge after 5 to 12 years [13].

The remaining 0.5 kg/m^2^ could be due to the service-associated individual nutrition, with physical inactivity and sedentary lifestyles likely typical of soldiers with prolonged service in the armed forces. In the civilian population, overweight and obesity are associated with emotional eating and stress. Soldiers are often exposed to special stress factors. An American study described an association between greater stress and a higher BMI and emotional eating. Stress thus influences BMI indirectly via emotional eating [14].

However, the increasing prevalence—shift from overweight to obesity (Table 5)—of 8.3% for those with 2–5 years of military service and of 15.1% for those with more than 5 years of service remains alarming, and each case should be avoided.

Our literature research revealed only one study that reported the BMI status or the prevalence of overweight and obesity from the entry to the release of the military service in Israel referring to a time span of 1989 to 2003 [15]. In a cohort of more than 22,000 soldiers—approximately 50% males and females—in the age of 20 to 22 years, the average increase in BMI during military service in Israel was 1.11 kg/m^2^ in males and 1.08 kg/m^2^ in females, which was quite similar to our observed average BMI gains of about 1 kg/m^2^, when the stated length of employment is not taken into account. The Israeli authors concluded that BMI appears to increase significantly in early adulthood, which is driven by the predictors of low individual and family educational attainment and, additionally, for females, oral contraceptive cessation as well as a low level of physical activity. Of those being normal weight at the entry of service, 12% became overweight during their army time, and of initially overweight individuals, 21.7% were obese at their release of service in the observation period between 1989 and 2003 [15]. In the Israeli conscription army, military service is three years for men and 20–21 months for women, followed by annually repeated service periods of one month up to the age of 43 years for males and 24 for females, respectively. The respective prevalence changes at the German Armed Forces (between 2010 to 2022) were 18.3% starting normal weighted and becoming overweight at their release (service time from 2–5 years with a median BMI increase of +1.1 kg/m^2^) or 6.4% (service time from under 2 years with a median BMI increase of +0 kg/m^2^), respectively. Compared to the Israeli 12%, the German trend in overweight prevalence was obviously also a function of length of service time, which became particularly clear for service lengths of more than five years (33.1% for change from normal to overweight, median BMI increase 2.4 kg/m^2^) taken into account. Changes in prevalence from overweight to obesity were considerably lower in the German Armed Forces (1.8% after less than 2 years, 8.3% after 2–5 years, 15.1% after more than 5 years) compared to 21.7% in the Israeli army.

Another main finding was that the prevalence of overweight soldiers increased from 33.0% at entry to 39.5% by the end of employment. The respective increases in obesity prevalence were 3.7% at entry and 6.3% at end of service again.

The BMI status assessed at the beginning of the service can be assumed to be relative to the situation in the general German population. Among applicants at entry into the German Armed Forces, BMI was about 23.5 kg/m^2^ on average. In the approximately age-adjusted German general population (data collected between 2008–2011), the mean BMI values were 24.5 kg/m^2^ for males and 23.7 kg/m^2^ for females, and the prevalence of overweight (BMI < 30 kg/m^2^) was 26.7% in men and 20.4% in women aged 18–29 years, while the prevalence of obesity (BMI ≥ 30 kg/m^2^) was 8.6% and 9.6%, respectively [1]. Although average BMI values in military applicants and the general population appear to be very similar, the prevalence of overweight may be unfavorable for prospective soldiers, while the prevalence for obesity was in favor of soldiers upon entry. When making comparisons, it should be taken into account that our data was assessed about a decade later than that of the general German population and that secular or contemporary effects are well known. In the Polish army, BMI assessments between the year 2000 and 2010 revealed a significant trend for an increasing prevalence of overweight from 10.5% to 15.6%, and of obesity from 2.5% to 3.8%, respectively [6]. Thus, in the Polish army, there was a 50% increase of prevalence within 10 years. In the Czech Army, during the monitored period over 11 years, from 1999 to 2009, the prevalence for overweight men was increasing from 52% to 57.1%. However, the prevalence of obesity among men has not changed significantly [16]. For the US Army, Hruby et al. reported a very similar trend between 1989 and 2012, with the prevalence increasing by about 50% for overweight—25.8% to 37.2%—and for obesity—5.6% to 8.0%—over a 23-year period on a markedly higher level [17]. Compared to the US Army, the BMI status among German soldiers was also favorable at the end of their military service. Possible explanations might be better nutrition advice, creating a diet plan together with physicians, more information about comorbidities or consequential damages, obesity counseling in military hospitals, and different offers for sports.

Some studies reporting BMI and the prevalence of overweight and obesity in the British, French, and Czech Armed Forces referred to an extended data collection during the soldiers’ service times [11,16,18]. These studies had access to a wider range of variables that predict an individual´s health status. This allowed them to perform regression analyses to predict health-relevant risk factors associated with the BMI and the WHtR, which was not applicable in our study because the German Armed Forces started regular health monitoring using the WHtR at least in the year 2018. However, BMI comparisons with our data set were feasible. A prevalence of 44.7% for overweight (BMI between 25 and 30 kg/m^2^) and of 12.0% for obesity (BMI ≥ 30 kg/m^2^) was reported for the UK Armed Forces, with males generally having a higher prevalence than females, which was not only higher than in our German Armed Forces cohort, but even higher than in the US Army [17]. In contrast, a lower prevalence of 38.7% in overweight and 10.0% in obesity with a mean BMI of 25.4 kg/m^2^ was found for the French Armed Forces from 2016 to 2017 [11]. Thus, the obesity prevalence was lower in the French army than in the British and Czech Armed Forces, but higher than in the German Forces, and so was the mean BMI.

Our study had strengths and limitations. One strength was definitely the large sample size and, in particular, the high number of paired cases that enabled us to consider more than 85 thousand individually paired datasets for analysis of BMI change. Another strength was the time span of more than a decade (2010 to 2022), which was helpful to account for likely nonspecific time effects also observed in the general population. A weakness to mention is that the original data set of almost half a million assessments had to be filtered because pair wise data comparisons were not available for each assessment. Another limitation was that we did not have access to interesting health-related socio-demographic data hampering a reasonable regression analysis of health risks. In this context, a French study was able to describe a correlation between obesity and socioeconomic status as well as educational level [11].

## 5. Conclusions

Our study demonstrated slightly increasing BMI values for soldiers entering the Bundeswehr over the past decade, which were in line with trends in the general population. However, BMI increases were higher for longer periods of service, especially for more than five years in the German Armed Forces, compared to age-adjusted residents in the general German population, which in turn must be taken into account for future perspectives on fitness and health behavior. We strongly recommend that the German Armed Forces adopt measures to improve fitness, e.g., additional offers and information on sports and exercise or even more offers and information on individual nutrition behavior.

## Figures and Tables

**Table 1 healthcare-11-00225-t001:** Frequencies for years of service described for one-year increments.

	Frequency	Percent
under 2 years	67,053	78.8
2 to 3 years	3780	4.4
3 to 4 years	3214	3.8
4 to 5 years	4471	5.3
5 to 6 years	910	1.1
6 to 7 years	809	1.0
7 to 8 years	1311	1.5
over 8 years	3528	4.1
total	85,076	100

**Table 2 healthcare-11-00225-t002:** Frequencies for clustered service duration classes.

	Frequency	Percent
under 2 years	67,053	78.8
2 to <5 years	11,465	13.5
5 to >12 years	6558	7.7
total	85,076	100

**Table 3 healthcare-11-00225-t003:** BMI state at the service entry (t1) and the release (t2) with BMI changes (Δ) for the total samples as well as males and females within service duration clusters (Median, IQR: 25–75%).

	Counts	BMI t1	25%	75%	BMI t2	25%	75%	Δt1 − t2	25%	75%
under 2 years	total (n = 67,033)	**23.4**	21.5	25.7	**23.8**	21.9	26.0	**0.0**	0.0	0.9
	females (n = 3231)	**22.7**	20.8	24.9	**22.9**	21.1	25.0	**0.0**	0.0	0.4
	males (n = 63,802)	**23.5**	21.5	25.8	**23.8**	22.0	26.0	**0.0**	0.0	0.9
2 to <5 years	total (n = 11,457)	**23.6**	21.6	26.0	**24.9**	22.9	27.4	**1.1**	0.0	2.7
	females (n = 853)	**22.5**	20.8	25.1	**23.5**	21.2	26.3	**0.7**	−0.2	2.2
	males (n = 10,604)	**23.7**	21.6	26.1	**25.0**	23.0	27.5	**1.1**	0.0	2.7
5 to >12 years	total (n = 6554)	**23.5**	21.6	25.7	**26.0**	23.9	28.7	**2.4**	0.6	4.4
	females (n = 463)	**22.8**	21.0	25.1	**24.5**	22.3	27.8	**1.6**	0.0	3.8
	males (n = 6091)	**23.6**	21.7	25.7	**26.1**	24.0	28.7	**2.5**	0.7	4.4

**Table 4 healthcare-11-00225-t004:** Overweight prevalence at the entry into and the end of employment depending on service duration (<2 years, 2–5 years, >5 years).

			Release from Service		
Service Duration		BMI	<25 kg/m^2^	≥25 kg/m^2^	Summed Up *n*
entry to service	<2 years (*n* = 67,019)	<25 kg/m^2^	40,879 (61.0%)	4281 (6.4%)	45,160
		≥25 kg/m^2^	2420 (3.6%)	19,439 (29.0%)	21,859
	2–5 years (*n* = 11,465)	<25 kg/m^2^	5313 (46.3%)	2098 (18.3%)	7411
		≥25 kg/m^2^	412 (3.6%)	3642 (31.8%)	4054
	>5 years (*n* = 6557)	<25 kg/m^2^	2197 (33.5%)	2169 (33.1%)	4366
		≥25 kg/m^2^	203 (3.1%)	1988 (30.3%)	2191

**Table 5 healthcare-11-00225-t005:** Obesity prevalence at the entry and at the release of employment depending on service duration (<2 years, 2–5 years, >5 years).

			Release from Service		
Service Duration		BMI	<30 kg/m^2^	≥30 kg/m^2^	Summed Up *n*
entry to service	<2 years (*n* = 67,019)	<30 kg/m^2^	63,284 (94.4%)	1222 (1.8%)	64,506
		≥30 kg/m^2^	722 (1.1%)	1791 (2.7%)	2513
	2–5 years (*n* = 11,465)	<30 kg/m^2^	10,086 (88.0%)	948 (8.3%)	11,034
		≥30 kg/m^2^	160 (1.4%)	271 (2.4%)	431
	>5 years (*n* = 6557)	<30 kg/m^2^	5386 (82.1%)	989 (15.1%)	6375
		≥30 kg/m^2^	49 (0.7%)	133 (2.0%)	182

## Data Availability

Not applicable.

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
