# Peer review of "BMI Alterations and Prevalence of Overweight and Obesity Related to Service Duration at the German Armed Forces"

_healthcare, 2023, doi:10.3390/healthcare11020225_

Round 1

Reviewer 1 Report

This descriptive retrospective cross-sectional study assessed 85,076 paired BMI data sets of German soldiers, between 2010 to 2022, using data provided by the Institute of Preventive Medicine of the German Armed Forces. As such, this work used a large sample size and its method/data analysis seems suitable. Although the manuscript is well written, the main objective of this study is not clear for the reader, because it is presented in different forms throughout the paper. Considering the purpose as a key point in the structure of the work, my first recommendation is to describe the aim of the study consistently across the text (i.e abstract, introduction and discussion). Particularly, I recommend to synthesize the aim of the study as something related to “comparison of BMI/weight status among different service time” or “to verify differences in BMI/weight status by service time clusters”. This is only a suggestion, but I really encourage the authors to reassess the formulation of the aim.

Furthermore, I have only few comments, as described per bellow

1) “…service duration had an impact on BMI increases.” (lines 22-23)

I think temerarious to consider as an “impact” given this study design. I suggest to use another term to describe the relation between service duration and BMI increases.

2) Considering the non-normal distributed data, what did you use Pearson correlation coefficient?

3) “…for continuous variables (e.g., employment duration, BMI alterations) we calculated the Pearson correlation coefficient (r).” (lines 125-126)

How exactly did you conduct correlation analysis? Did you use measures of both time points (pared measures)? If yes, again, why did you use Pearson correlation? Please, clarify!

4) “This ranking was associated with a significant correlation between years of service and BMI change within the whole sample of 85,076 paired data sets (r=0.35, p<.001)” (lines 183-184)

Is BMI change the difference (delta) between BMI T1 and T2? If yes, please clarify in the data analysis and in the Results. If not, please clarify these findings.

5) I suggest to join the first and second paragraphs of the Discussion as a single.

6) “In conclusion, the obesity prevalence and mean BMI were lower in the German Armed Forces than in the US, British, Czech and French Army.” (339-340)

I recommend to withdraw this conclusion, because it gives the idea that the comparison between German and other Armed Forces was done in this study.

Author Response

This descriptive retrospective cross-sectional study assessed 85,076 paired BMI data sets of German soldiers, between 2010 to 2022, using data provided by the Institute of Preventive Medicine of the German Armed Forces. As such, this work used a large sample size and its method/data analysis seems suitable. Although the manuscript is well written, the main objective of this study is not clear for the reader, because it is presented in different forms throughout the paper.

Considering the purpose as a key point in the structure of the work, my first recommendation is to describe the aim of the study consistently across the text (i.e abstract, introduction and discussion).

Particularly, I recommend to synthesize the aim of the study as something related to “comparison of BMI/weight status among different service time” or “to verify differences in BMI/weight status by service time clusters”. This is only a suggestion, but I really encourage the authors to reassess the formulation of the aim.

Answer: Thank you very much. We will address your suggestion and will find a consistent phrasing for the study purpose throughout the manuscript at all stages: abstract – introduction – discussion with your suggestion of ‘The purpose was to verify differences of BMI status increases by service time clusters.

Furthermore, I have only few comments, as described per bellow

 1) “…service duration had an impact on BMI increases.” (lines 22-23)

I think temerarious to consider as an “impact” given this study design. I suggest to use another term to describe the relation between service duration and BMI increases.

Answer: Thank you. We will rephrase this sentence: “…service duration was related to BMI increases”.

 2) Considering the non-normal distributed data, what did you use Pearson correlation coefficient?

Answer: Please excuse me, but the Pearson correlation only needs a linear relation and interval scaled data, but not necessarily normally distributed data sets, but we can calculate the Spearman coefficient as well. I guess you are right and that will be the more appropriate measure, thank you.

Spearman rho rs = 0.337 (p<.001)

3) “…for continuous variables (e.g., employment duration, BMI alterations) we calculated the Pearson correlation coefficient (r).” (lines 125-126)

How exactly did you conduct correlation analysis? Did you use measures of both time points (pared measures)? If yes, again, why did you use Pearson correlation? Please, clarify!

Answer: Sorry, I have to apologize. Obviously the idea was not quite clear. We calculated correlation analyses only between the variables service duration (years) and BMI alterations (differences between t1 [start of service] and t2 [end of service] referring to the whole sample of more than 85 thousand persons with paired data sets.

 4) “This ranking was associated with a significant correlation between years of service and BMI change within the whole sample of 85,076 paired data sets (r=0.35, p<.001)” (lines 183-184)

Is BMI change the difference (delta) between BMI T1 and T2? If yes, please clarify in the data analysis and in the Results. If not, please clarify these findings.

We addressed this concern in the paragraph above. We will change the Pearson correlation (r) into the Spearman correlation (rs) coefficient in the methods’ section and will clarify in the results’ section that the correlation analysis referred to BMI changes (delta t1-t2) with service duration. The sentence will be then:

“This ranking was associated with a significant correlation between years of service and BMI changes within the whole sample of 85,076 paired data sets (r=0.34, p<.001)” (lines 183-184) 

And ‘yes’ BMI changes mean the difference (delta) between t1 and t2, as explained in Table 4.

5) I suggest to join the first and second paragraphs of the Discussion as a single.

Answer:  OK – will be joined. Thank you.

6) “In conclusion, the obesity prevalence and mean BMI were lower in the German Armed Forces than in the US, British, Czech and French Army.” (339-340)

I recommend to withdraw this conclusion, because it gives the idea that the comparison between German and other Armed Forces was done in this study.

Answer: Thank you. You are right. We did not assess other data than those of the German Armed Forces. The sentence will be withdrawn from the conclusions.

Reviewer 2 Report

The manuscript titled" BMI alterations and prevalence of overweight and obesity-related to service duration at the German Armed Forces by Lorenz Scheit et al. is comprehensive and well-written. However, it needs improvement.
  1. In table 4, the authors have described the BMI (kg/m2) at the entry to service for the total sample from the year 2010 to 2022 (Median, 159 whiskers: IQR 25%–75%) wonderfully, but how have they calculated the t1-t1,25% and 75% in last three columns. Please describe it in detail.
  2. The sample size for the 5 to 12 years group is too small. The authors should enhance the sample size to the made conclusion.
  3. The authors should add the physical activity or duration of exercise for all groups.

Author Response

  1. In table 4, the authors have described the BMI (kg/m2) at the entry to service for the total sample from the year 2010 to 2022 (Median, 159 whiskers: IQR 25%–75%) wonderfully, but how have they calculated the t1-t1,25% and 75% in last three columns. Please describe it in detail.

Answer: Dear reviewer, I am not quite sure, whether I understand your question correcly ? As our analyses were based on paired data sets (BMI at entry [t1] and exit [t2] from service), we were able to calculate the individuals’ changes from t1 to t2. And these changes were the basis for the descriptive statistics (median, 25% and 75% quartile) not only for the single time points but also for the ‘changes’ columns. Maybe, you were wondering about the median = 0, and the 25% quartile to be either 0 or even negative in one case?

I think you will understand better, when you take into account the results given under the sub-heading 3.2, which explain that there were BMI maintainers and decreases, but a majority demonstrated increases:

“In detail, a number of 26,416 soldiers showed no BMI changes, while a minority of 18,462 soldiers showed BMI decreases, and a majority of 40,151 soldiers showed BMI increases”.

  1. The sample size for the 5 to 12 years group is too small. The authors should enhance the sample size to the made conclusion.

Dear reviewer,

An extension of the sampel size in the period 5 - 12 years is not possible, because the data available here represents all available case numbers. The specific personnel structure of the German Armed Forces is such that the number of individuals decreases as the service period increases, hence the relatively smaller number of data sets for this period compared to the shorter periods. Therefore, the sample size can´t become enhanced technically. Nevertheless the total sample size is > 85 thousand and it the sampel size (5 – 12 years) of n = 6557 seems not a small number, especially. Of course, the group with a service duration of  >5 years has a percentage of only 7.7% (table 2), but the absolute number of >6,500 is anything else than negligible – it should be rated as a huge number of cases, I suggest.

  1. The authors should add the physical activity or duration of exercise for all groups.

Answer: Dear reviewer,  

Thank you for this suggestion and we would like to add the information of physical activity, but the Institute of preventive Medicine of the German Armed Forces wasn´t able to deliver more interesting materials.  We mentioned in the methods section, that we unfortunately had no access to other personalized data than the date of service entry/ release and the body mass and height at the respective time points. Based on these data, we calculated the soldiers’ age and their BMI, and in a second slope, we determined the state of normal or overweight or obesity. Other information may possibly be obtained in future studies. 

Reviewer 3 Report

1) Title & abstract correspond to each other. 

2) Introduction provided a good background to the topic, highlighted the research gap and gave a fair idea of the study aims. 

3) Methods

2.1 - authors have mentioned all possible observational study designs here. Please decide on one design. 

2.2 - the content here not reflecting "setting" of the study. Please choose a more appropriate title.

Line 123 - the term "correlation" is not accurate when it comes to categorical variables. 

I believe for this sample size, authors could have done a multivariate analysis to control for confounding factors, as the IV of interest is the service duration only. Please explain why this is not done. 

4) Results

Table 3 and Figure 1 are redundant. 

Table 4 - dots (.) appeared as comma (,)

Table 4 and Figure 2 are redundant. 

Line 200 - 221 provides all of info about chi square results and cited Table 5 & 6. However, these figures are not available in the cited Tables, which were merely descriptive tables. I also fail to see Pearson's r in any tables. 

5) Detailed discussion has been provided. 

Author Response

1) Title & abstract correspond to each other. 

2) Introduction provided a good background to the topic, highlighted the research gap and gave a fair idea of the study aims. 

3) Methods

2.1 - authors have mentioned all possible observational study designs here. Please decide on one design. 

Answer: Thank you for this advice. In fact, our chi²-based analyses of the prevalence of overweight or obesity refer to a cohort study design. A little bit unusual for a cohort study is the fact that we did not only count presence or absence of a variable status. We also compared individually paired data sets of the same individuals under the condition ‘at entry’ or ‘at discharge’ of service. But actually, we decided to rephrase as follows: “The present study analyzed data from the years 2010 to 2022 in terms of a descriptive retrospective cross-sectional cohort study”.

2.2 - the content here not reflecting "setting" of the study. Please choose a more appropriate title.

Ansewr: Thank you very much again. You are right here. We chose to find a more appropriate sub-heading: “Data acquisition and preparation” (lines 85-86).

Line 123 - the term "correlation" is not accurate when it comes to categorical variables. 

I believe for this sample size, authors could have done a multivariate analysis to control for confounding factors, as the IV of interest is the service duration only. Please explain why this is not done. 

Answer: Thank you. We changed the formulation in line 123:

For contingency coefficient in the case of correlation analyses between categorical variables (e.g. overweight or obesity status at entry or end of service) we calculated chi²-based McNemar tests and the contingency coefficient (C), and for continuous variables (e.g., employment duration, BMI alterations), we calculated the Spearman correlation coefficient (rs).

4) Results

Table 3 and Figure 1 are redundant. 

Sorry, I have to give a short rebuttal. You are right that table 3 and fig. 1 mirror the same information, but the information itself is necessary, because we have to show the tendency of a slightly increasing BMI at service entry in order to discuss any secular effects over the observational decade to interpret correctly the BMI status and corresponding increases during the service durations. To my opinion, the table 3 gives more exact information, but the illustration (fig. 1) is easier and quicker to perceive. If possible, I would be happy, if both the table and the figure could be remaining.

Table 4 - dots (.) appeared as comma (,)

Answer: Sorry. You are right. We substituted the (,) by the (.). Thank you.

Table 4 and Figure 2 are redundant. 

Answer: We deleted Figure 2

Line 200 - 221 provides all of info about chi square results and cited Table 5 & 6. However, these figures are not available in the cited Tables, which were merely descriptive tables. I also fail to see Pearson's r in any tables.

Answer: Dear reviewer, I have to give a short rebuttal again. Yes of course, the tables are descriptive. The chi²-based significance tests and p-values have to be accompanied by the respective cross tabs, which demonstrate the absolute numbers and the percentages – especially these percentages were needed for the discussion. Only reporting the p-values would not be enough, I suppose. The table contains necessary information and provides an overview for the data mentioned in the text flow.

The correlation coefficient (we followed another reviewer’s recommendation and switched to Spearman’s rho) does not appear in any table, because it was calculated based on the whole sample (>85 thousand) and was not related to sub-group analyses given in the tables 4-6. 

5) Detailed discussion has been provided. 

Thank you.

Reviewer 4 Report

The current manuscript analyzed the BMI data from 2010 to 2022 of soldiers from the German Armed Forces. Their retrospective cross-sectional study seems to suggest an increase in BMI alterations with the length of service. Overall, the manuscript is well-organized and well-written. However, some important factors were neglected as the authors mentioned in the discussion, such as diet patterns, activity time, lifestyles, and which military part of the subject was on duty. The conclusion lacks some convincingness without taking those factors into consideration.

Author Response

The current manuscript analyzed the BMI data from 2010 to 2022 of soldiers from the German Armed Forces. Their retrospective cross-sectional study seems to suggest an increase in BMI alterations with the length of service. Overall, the manuscript is well-organized and well-written. However, some important factors were neglected as the authors mentioned in the discussion, such as diet patterns, activity time, lifestyles, and which military part of the subject was on duty.

The conclusion lacks some convincingness without taking those factors into consideration.

Answer: Dear reviewer, thank you very much for your comment. Following another reviewer’s recommendation, we deleted the last sentence of the conclusions paragraph, because in the conclusion’s section only results of our own data assessment and corresponding analyses should be considered.

Thus, we removed that BMI increases were less compared to other Armed Forces. As we had no access to any other data than age, body mass and body height at the respective time points at service entry and discharge, our conclusions should focus exclusively on these parameters.

But in the frame work of the discussion, we undertook comparisons with other articles showing both BMI Status analyses and additional information about health related variables like diet or physical activity or fitness levels for which we had unfortunately no access.

To be honest, I would be happy, if I could convince you that other than the analyzed data should not be part of the conclusion. Thank you.

Reviewer 5 Report

Dear Editors,

Thank you very much for considering me for the review of this paper. Many thanks to the authors for their time in preparing this interesting manuscript.
The following is a series of contributions with the intention of being able to contribute to the paper if the authors deem it appropriate.

In the abstract, a multitude of data are provided and should not be provided. Only highlight the most important result or results. Numbers should not appear. Neither in the sample where they only refer to the % of women and men, do not exist?, when it can be deduced that they are the majority in this study.
In the introduction, there is a review of the state of the art which is not very good because there are not many studies. Regarding the objective or objectives, which are not very clear, the authors could make a seriation, to highlight them clearly.
In the design of the study, the authors indicate that the level of evidence obtained is 3. What type of scale is it? On the basis of which author or authors?

Section 2.2. the authors should divide it into a section on the description of the sample and a section on the data collection procedure. That way it would be much clearer.
Regarding the statistical analysis, how did the authors know that it is a non-normal distribution? They should specifically indicate the statistical tests right at the beginning of the section and not later; the result of this test determines the type of tests to be used.

Section 3.1. corresponds to the description of the sample.
Regarding the rest of the results, the authors should not repeat the data appearing in the tables. Many are repeated, mainly in Table 5.
The discussion is very deficient. The authors repeat data that appear in the results and, on the other hand, the presence of current studies is very scarce.
The literature review is very scarce. There are very current works on the subject that demonstrate the existence of important studies that the authors have not considered. For example, the study cited below:

Fernández-Donet, R.; Marco-Ahulló, A.; Bermejo, J. L.; Monfort-Torres, G. (2021). Epidemiología y principales causas de lesión de militares pertenecientes al ejército de tierra. Journal of Sport and Health Research. 13(1): 93-102

It is recommended that the paper be thoroughly revised in order to achieve the quality required for a scientific article of the level of this journal.

Author Response

In the abstract, a multitude of data are provided and should not be provided. Only highlight the most important result or results. Numbers should not appear. Neither in the sample where they only refer to the % of women and men, do not exist?, when it can be deduced that they are the majority in this study.

Answer: Dear reviewer, thank you for your suggestion, but I have to give a short rebuttal here. I suppose, there might be differing attitude towards ‘numbers’ in the abstract. But we – as well as the other 4 reviewers – found the information given in the abstract was suitable. And actually, the title of our manuscript refers to BMI status and overweight and obesity alterations being related to service durations, and in the abstract’s results we focused on a few numbers referring to BMI increases and three duration clusters and the respective correlation with additional information of obesity prevalence changes – these are central results. Moreover, it is assumed to be a valuable information that our findings were based on a large scaled sample size (>85 thousand) with a minor proportion of females (5.4%), which appears not to be redundant information. I hope that you might agree to this point of view.

In the introduction, there is a review of the state of the art which is not very good because there are not many studies. Regarding the objective or objectives, which are not very clear, the authors could make a seriation, to highlight them clearly.

Answer: Dear reviewer, we made some rephrasing to clarify better the study goals or objective. Referring to your comment on the number of references in the introduction, we do not agree. In line with the other 4 reviewers, we think that we provided necessary back ground information in order to highlight the present research gap. And – here you are right – we made some changes for the phrasing of the study goals. Thank you very much.

In the design of the study, the authors indicate that the level of evidence obtained is 3. What type of scale is it? On the basis of which author or authors?

Answer: Our study has no experimental study design, but it refers to a very large scaled sample (cohort), and we conducted correlation and contingency analyses (r, C/chi²). Thus, it is descriptive and correlative, which is much more than an opinion (level 4). Therefore we guess, the note of evidence level 3 would be appropriate, if the journal wishes any note, here.

Section 2.2. the authors should divide it into a section on the description of the sample and a section on the data collection procedure. That way it would be much clearer.

Answer: Actually, we structured our Section 2.2 (re-named: data acquisition and preparation) in a way that informed the reader what parameters were delivered from the data base, and what kind of filters had to be applied in order to enable us to analyze pre-post (entry-exit) BMI status alterations. This should be transparent. Maybe, I misunderstood your recommendation?

Regarding the statistical analysis, how did the authors know that it is a non-normal distribution? They should specifically indicate the statistical tests right at the beginning of the section and not later; the result of this test determines the type of tests to be used.

Answer: Thank you. We inserted a respective sentence. We tested for normality (Kolmogorof-Smirnof test) and found significant differences between normal distribution and the distribution of our data. But for our outcome variables (age, BMI) of soldiers at their entry or discharge of service, it is well known that normality is seldom found – parametric statistics were showing standard deviations for age and BMI that were valid (age under 18 years, BMI in the range of underweight). Skewed distributions demonstrating more variation above than under the mean (median) are the regularly observed distributions.

Section 3.1. corresponds to the description of the sample.
Regarding the rest of the results, the authors should not repeat the data appearing in the tables. Many are repeated, mainly in Table 5.

Answer: There are differing opinions towards the degree of reasonable repetitions. Mostly, we reported main results and then we refer to a corresponding table, which gives more detailed information (e.g. information text and table 3 plus fig. 1, text and table 4 plus fig. 2). But actually the chi²-based significance testing results had to be accompanied by the referring numbers (overweight or obesity prevalence) highlighted in the text flow and given completely in the table 5 and 6. There is no redundancy, from our point of view. The numbers given in the text represent an exclusive choice of the whole set of numbers shown in the tables.

The discussion is very deficient. The authors repeat data that appear in the results and, on the other hand, the presence of current studies is very scarce.

Answer: We think, the discussion is meant to repeat study goals and central results, which in turn have to be named and then interpreted facing the current literature. Maybe we did not detect some relevant reference, although we started our literature research carefully on “BMI status and changes related to service time at the Armed Forces”. We have to apologize.

The literature review is very scarce. There are very current works on the subject that demonstrate the existence of important studies that the authors have not considered. For example, the study cited below:

Fernández-Donet, R.; Marco-Ahulló, A.; Bermejo, J. L.; Monfort-Torres, G. (2021). Epidemiología y principales causas de lesión de militares pertenecientes al ejército de tierra. Journal of Sport and Health Research. 13(1): 93-102

Answer: I am sorry, but your proposed article is absolutely off-topic. It has nothing to do with our study aiming to investigate BMI status and changes during differing service time clusters at the German Armed Forces.

Objective. To make an approximation to the epidemiology and the main causes of injury in the Spanish Army. Methods. The sample consisted of 130 soldiers belonging to the Spanish Army, who completed a questionnaire relating to issues such as the history of injuries, physical preparation, sports habits and personal issues. A correlational analysis was made by means of the Pearson test among the inventoried variables. Results. 56.7% of those surveyed stated that they had had injuries during their working hours, with the lower limbs being the areas most affected by injuries (70%) and ligament injuries the most frequent (41.8%). On the other hand, running was the practice that triggered the greatest number of injuries. Injuries occurred during work maintained a moderate relationship with physical activity performed outside of work, and negative with hydration. Discussion. Injuries represent a major problem for the military community. These data indicate the high level of physical demands to which the military are subjected, fortunately, the reviewed scientific literature as well as the work done in this article indicate that guidelines can be created for their avoidance. Conclusions. The results corroborate a high injury rate during working hours, which represents a problem for the military community. The results obtained by this work can be helpful in creating guidelines to reduce this high number of injuries. ABSTRACT FROM AUTHOR

It is recommended that the paper be thoroughly revised in order to achieve the quality required for a scientific article of the level of this journal.

Answer: We are very sorry that you are not satisfied with our manuscript. Some of your recommendations appear to be helpful, others do not, I am afraid. We will try to make some changes in order to improve our manuscript.

Round 2

Reviewer 2 Report

In this revised manuscript, the authors have addressed all of the concerns raised by the reviewers. I believe that this manuscript is now ready for acceptance. 

Author Response

Dear reviewer 2,

thank you very much for your constructive help and the acceptance of the manuscript.

Reviewer 3 Report

The most important issue here is that I think the authors have misunderstood my comment on the need for multivariate analysis, as correlations etc are not multivariate analysis which could have authors control for confounders. Hence, this issue remains unaddressed. 

I have to disagree with the rebuttal with regard to Table 3 & Figure 1. It's not a standard practice to have the same data presented in different forms within the manuscript. Should the authors feel it's necessary to show the exact numbers, it can be a supplementary table but not disrupt the flow of the manuscript. 

Accurate results must be presented in Tables. This is in reference to the correlations - that should be presented in the Tables and only important results highlighted in the text. 

Author Response

Rev#3: issue „multivariate analyses“

Dear Reviewer,

You are right; we did not address your concern about probable multivariate analyses. Maybe, we misunderstood each other hitherto…

If I understand right, you propose a regression analysis, and not a simple bivariate correlation?

The problem is that we did not have access to any interesting variables being able to explain the BMI increases except for sex and age.

I will insert here a calculation (SPSS: linear regression, fixed effects, standardized β-coefficients) using BMI increases as the criterion and the predictors ‘age at release of service =age_t2’ and ‘service duration’ and ‘sex’:

R

R-square

corr R-square

SEE

0.446

.199

.199

1.9910

non-standardized coefficent

standardized coefficient

T

Sig.

95,0% confidence interval for B

regression coefficient

SE

Beta

lower

upper

age_release

.002

.000

.018

4.186

.000

.001

.002

service duration

.357

.003

.436

105.032

0.000

.351

.364

sex

-.162

.030

-.017

-5.337

.000

-.221

-.102

BMI-delta

BMI-delta

1.000

age_release

.307

service duration

.445

sex

.061

And the age is a direct function of the service time elapsed from entry to release of service. Moreover, our sample of more than 85 thousand persons consisted in the absolute majority of males. There were only about 5% females.

I am not sure, if it is a valid approach to come to a higher ß-coefficient (0.436) instead of a Spearman rho = 0.34 for a dependency between BMI increases and service duration by adding some other variables (sex with only 5% participants, or age being directly dependent on service duration).

By the way, following the recommendations of another reviewer, we switch from the parametric Pearson correlation to the non-parametric Spearman rho coefficient. Conducting a multivariate regression would lead to additional problems here…

But of course we would have appreciated additional information like the rank of the soldiers (officer vs. regular soldier), or ‘combat unit’ vs. ‘administration, maintenance repair, military staff’, or data describing ‘physical activity/ fitness’, or even ‘health problems ‘.

But unfortunately, we did not have access to any of these variables; to our knowledge, our work is the very first trying to evaluate BMI data in such a large scaled sample for the German Armed Forces (Bundeswehr).

Probably, future investigations are able to get access to more detailed data base information. But actually, Bundeswehr privacy rules hamper our investigations.

Does this answer address your concerns?

Reviewer 5 Report

Dear editors and authors,

Thank you very much for your responses and the time you have taken to do so.
Since you have barely accepted any of my suggestions I am not going to waste your time with any more or enter into discussion on those that have not been accepted. I don't know if my suggestions were not accepted or not understood.
In the abstract, I intended to indicate that if the % of women contributed it should also be men. However, I think that no % should be provided in order to invite the reader to review the paper. It´s an opinion, not a rule.

I do not know if they have accepted only one suggestion regarding the statistical test, and the authors have written it with spelling mistakes. It should be written Kolmogorov-Smirnov and not as the authors do, KolmogoroF-SminoF.

They have not accepted the suggestions on bibliographic review and we are faced with a study with only 18 references, which have errors according to the referencing rules of the journal. Considering that the authors have done a good review, perhaps the journal should consider whether this is a topic of interest to science or whether it is the right journal for a publication on the armed forces. I am afraid that this paper is not going to receive many citations.

I wish the authors every success and leave it to the journal to decide.
Kind regards

Author Response

Dear Reviewer 5

We would like to thank you very much for the constructive criticism, your time and your opinion.  It is nothing unusual that different opinions can arise in scientific evaluations.

We consider the indication of the percentage for women in the abstract to be quite reasonable and legitimate; in particular, an additional mention of the percentage for men is not necessary, as it follows automatically from a purely logical point of view.

We have corrected our spelling mistake for Kolmogorof-Sminof, thank you for bringing this spelling mistake to our attention.

The quality of an article is not measured by the quantitative number of cited references, but by the significance. Based on a thorough literature search, all articles relevant to this issue have been included. Many very good articles in the Journal Health Care have a similarly high number of references.

Two errors regarding the review rules in the literature citation have been corrected, thank you for pointing them out.

Both the topic of obesity and the military component are frequent and cited topics. Due to the fact that with 85,000 data sets a very high significance is available and that this article describes the connection between overweight and service time in the German military for the first time, I assume that the article should be of relevant scientific interest.

The authors concurred on many correction aspects with reviewers 1-4, who in turn did not share the aspects you raised. Of course, the editors of the journal will make their own judgment and decide about a possible publication.

Many thanks to all involved for their helpful support
